# Consistency of superb microvascular imaging and contrast-enhanced ultrasonography in detection of intraplaque neovascularization: A meta-analysis

**Fang Yang, Cong Wang** *

Department of Ultrasound, the First Affiliated Hospital of Dalian Medical University, Dalian, Liaoning Province, China

* wc027214@163.com

## Abstract

This meta-analysis assesses the consistency of superb microvascular imaging (SMI) and contrast-enhanced ultrasonography (CEUS) in detecting intraplaque neovascularization (IPN). We searched PubMed, Web of Science, the Cochrane Library, and CBM databases. A meta-analysis was conducted using STATA version 15.1 software. We calculated the pooled Kappa index. Ten studies that met all of the inclusion criteria were included in this meta-analysis. A total of 608 carotid plaques were assessed through both SMI and CEUS. The pooled summary Kappa index was 0.743 (95% confidence interval (CI) = 0.696–0.790) with statistical significance ($z = 31.14$, $P < 0.01$). We found no evidence of publication bias ($t = −1.21$, $P = 0.261$). Our meta-analysis indicates that SMI and CEUS display a good consistency in detecting the IPN of carotid plaque; that is, SMI ultrasound may be a promising alternative to CEUS for detecting the IPN of carotid plaque.

## Introduction

In today's society, the incidence of atherosclerosis is high, and it is trending younger [1]. Atherosclerotic plaques can cause carotid artery stenosis and affect the blood supply to the brain from the carotid artery to the brain, and vulnerable carotid atherosclerotic plaques are prone to rupture, bleeding, and the formation of thrombi, which can enter the blood vessels of the brain with the blood, causing ischemic stroke events [2]. Stroke is a common refractory disease that seriously endangers human health and lives [3]. The development of atherosclerotic plaque seriously affects the outcome and prognosis of the disease, and there is a significant consistency between intraplaque neovascularization (IPN) and atherosclerotic plaque vulnerability, so the IPN can be used as a risk factor to evaluate the vulnerability of plaque [4]. Contrast-enhanced ultrasonography (CEUS) can indicate IPN effectively, but it is an invasive examination requiring injection of a contrast medium [5]. Superb microvascular imaging (SMI) is a new ultrasonic diagnosis technology that uses adaptive principles to display low-speed blood flow signals [6]. Several studies had suggested that SMI, which is a promising noninvasive

**Data Availability Statement:** All relevant data are within the manuscript and its Supporting Information files.

**Funding:** The author(s) received no specific funding for this work.

**Competing interests:** The authors have declared that no competing interests exist.

alternative, can be used to detect IPN with accuracy comparable to CEUS [7]. However, the results of these studies have been contradictory and the sample sizes were small. Therefore, we performed the present meta-analysis to assess the consistency of SMI and CEUS in detecting intraplaque IPN.

## Methods

### Literature search

We searched PubMed, Web of Science, the Cochrane Library, and CBM databases. The following search terms were used: [carotid] and [plaques or plaque or fatty streak or fibroatheroma] and [contrast-enhanced ultrasound or contrast-enhanced ultrasonography or contrast ultrasonography or ultrasound contrast imaging or CEUS] and [vulnerability or stability or neovascularization] and [superb microvascular imaging]. We also reviewed references from eligible articles for additional relevant studies.

### Selection criteria

The eligible studies were required to match all the following criteria: (1) the study design must be a clinical cohort study, (2) the study must relate to the comparison of CEUS and SMI for detecting IPN, (3) intraplaque microvascular flow (IMVF) were be graded, and (4) published data in the row x column tables must be sufficient for Kappa index and standard error. In cases of sequential and duplicate publications, we included the most recent work with the largest sample size.

### Data extraction

The following data were extracted from each included study by two reviewers independently: year of article, the first author's surname, sample size, number of IMVF grades, Kappa index, and standard error.

### Quality assessment

Methodological quality was independently assessed by two researchers using a tool for the assessment of the quality of methodological indexes for nonrandomized studies (MINORS). The MINORS criteria included 12 assessment items. Each of these items was scored as "yes" (2), "no" (0), or "unclear" (1). MINORS score ranged from 0 to 24; and a score $\geq 17$ indicated good quality.

### Statistical analysis

The software STATA version 15.1 (Stata Corporation, College Station, Texas, USA) was used for meta-analysis. We calculated the pooled summary Kappa index and its 95% confidence interval (CI). The Cochran's Q-statistic and $I^2$ test were used to evaluate potential heterogeneity between studies [8]. If a Q test showed a $P < 0.05$ or $I^2$ test exceeded 50%, which indicates significant heterogeneity, the random effect model or fixed effects model was used. Sensitivity analysis was performed to evaluate the influence of each individual study on the overall estimate. The Begg's funnel plot and Egger's test were used to assess publication bias [9].

Table 1. Baseline characteristics and methodological quality of all included studies.

| First Author | Year | Sample Size | Gender (M/F) | Age (Years) | IMVF Grade Number | Kappa Index | SE (Kappa) | MINORS Score |
|---|---|---|---|---|---|---|---|---|
| Ma et al. [10] | 2018 | 55 | 34/12 | 61±7 | 4 | 0.734 | 0.072 | 18 |
| Chen [11] | 2016 | 80 | 40/16 | 64.1±8.2 | 4 | 0.755 | 0.059 | 19 |
| Zhang et al. [12] | 2019 | 60 | 29/14 | 62.9±7.0 | 4 | 0.650 | 0.076 | 18 |
| Ding et al. [13] | 2019 | 62 | 50/12 | 61.59±9.16 | 3 | 0.769 | 0.068 | 18 |
| Cheng et al. [14] | 2015 | 57 | 44/13 | 61.8±7.8 | 3 | 0.607 | 0.127 | 19 |
| Dong et al. [15] | 2018 | 69 | 39/30 | 67.38±8.61 | 3 | 0.689 | 0.105 | 17 |
| Zhang et al. [16] | 2017 | 39 | 27/12 | 60±4 | 3 | 0.860 | 0.076 | 17 |
| Yan [17] | 2018 | 33 | —— | —— | 4 | 0.621 | 0.103 | 19 |
| Xie et al. [18] | 2018 | 108 | 53/16 | 68.1±8.8 | 4 | 0.748 | 0.054 | 20 |
| Wang et al. [19] | 2019 | 45 | 33/12 | 61.42±7.37 | 4 | 0.839 | 0.090 | 18 |

M, male; F, female; IMVF, intraplaque microvascular flow; SE, standard error; MINORS, methodological index for nonrandomized studies.Quantitative data synthesis

## Results

### Characteristics of included studies

Overall, 10 studies (Table 1) that met all of the inclusion criteria were included in this meta-analysis. Fig 1 shows the selection process of the eligible articles. A total of 608 carotid plaques were assessed through both SMI and CEUS. MINORS scores of all the included studies were 17.

### Quantitative data synthesis

The fixed effects model was used whenever there was a lack of obvious heterogeneity among the studies ($I^2$ = 0%, $P$ = 0.531). Sensitivity analysis was carried out, and none of them caused obvious interference to the results of this meta-analysis (Fig 2). The pooled summary Kappa index was 0.743 (95% CI = 0.696−0.790) with statistical significance (z = 31.14, $P$ < 0.01), which indicated that SMI and CEUS have a good consistency in detecting IPN of carotid plaque (Fig 3). The funnel plots indicated little evidence of significant publication bias (Fig 4), and Egger's test confirmed this (t = 1.21, $P$ = 0.261).

## Discussion

Carotid atherosclerotic plaque is closely related to cerebrovascular events. It directly affects structure and composition of plaque, which indicates the occurrence and development of subsequent ischemic cerebrovascular events [20]. Those unstable plaques that readily rupture, fall off, and cause distal embolism are called vulnerable plaques [21]. The morphological characteristics of vulnerable plaques include the following: irregular plaque surface or ulcer formation, thin fiber cap or fissure, bleeding, high concentrations of lipids and inflammatory active components, and neovascularization in the plaque [22]. Neovascularization plays a central role in plaque initiation, progression, and rupture and is a predictor of plaque instability and stroke risk [23]. Histopathological evidence shows that, unlike with relatively stable plaques, the presence and density distribution of neovascularization in plaques are closely related to plaque rupture, and neovascularization is often located in the fibrous cap fissure area and in areas with lipid enrichment and active inflammation. The detection of neovascularization in plaque can be used to evaluate the stability of atherosclerotic plaque and even predict the occurrence of cardiovascular and cerebrovascular diseases [24]. Finding a convenient, safe, and

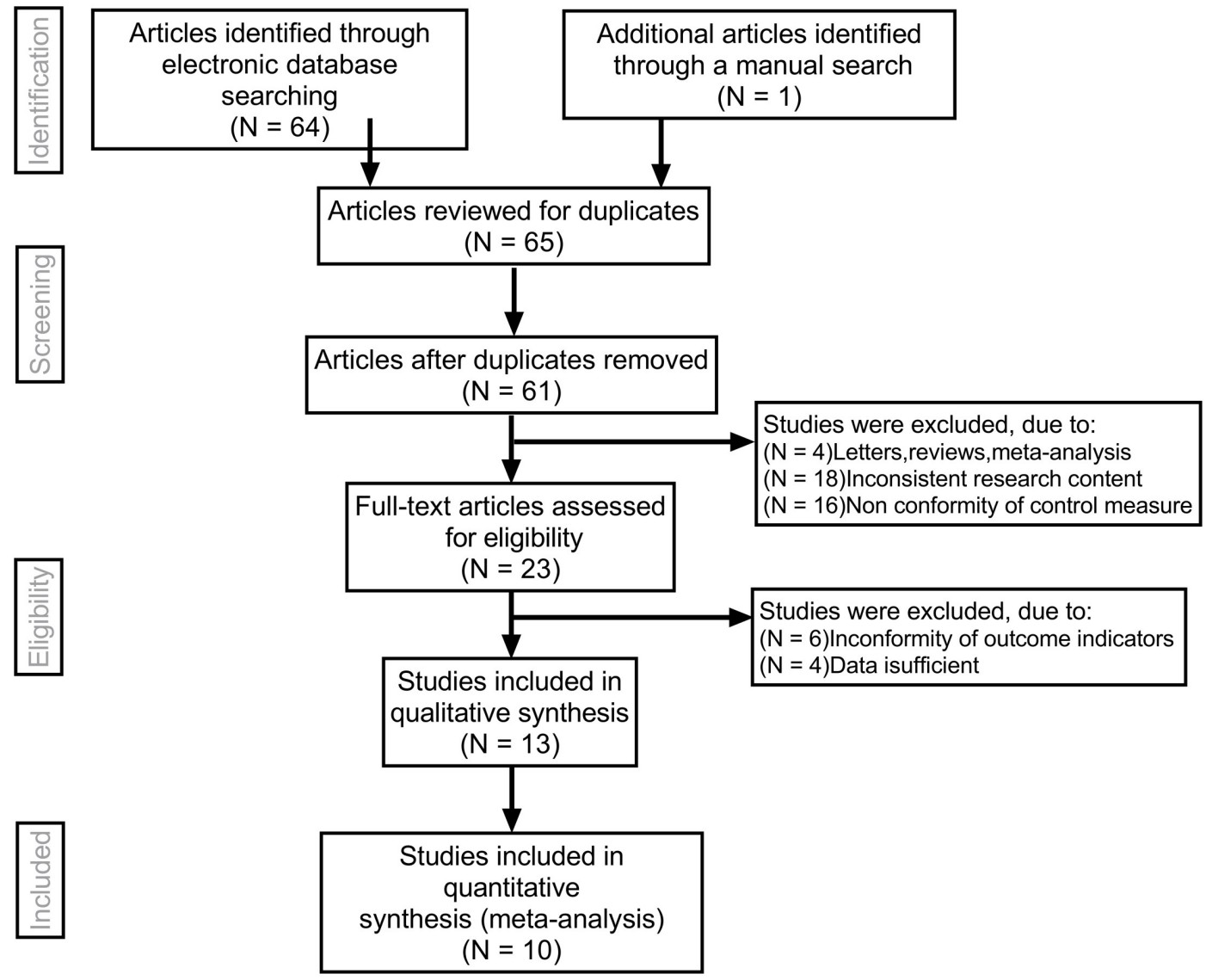

**Fig 1. Flow chart of literature search and study selection.** Ten studies were included in this meta-analysis.

reproducible imaging method that could be used to determine the stability of arterial plaque has long been the focus of clinical research.

Conventional two-dimensional ultrasound and color Doppler ultrasound can be used to observe and measure the echo, shape, and thickness of a plaque and evaluate the degree of vascular stenosis caused by plaque, but they cannot evaluate the stability of plaque comprehensively and accurately, whereas CEUS has high spatial and temporal resolution, and microbubbles have the same fluidity as red blood cells. Some scholars use CEUS to detect neovascularization in plaque as reliable evidence for the diagnosis of vulnerable plaque [25, 26]. However, because of the high cost of contrast media, trauma examination, and the risk of allergy of contrast media, CEUS is limited to some extent. It is necessary to find a simple, noninvasive, and inexpensive method of ultrasound examination.

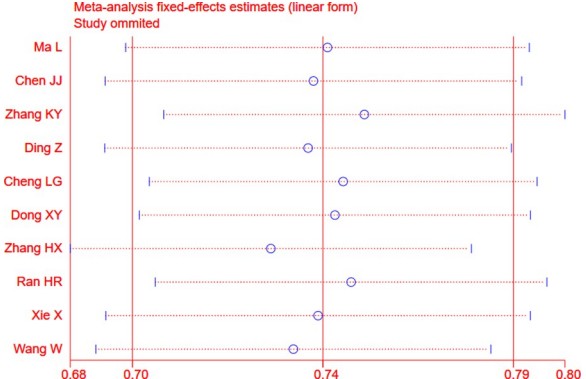

**Fig 2. Sensitivity analysis.** No one factor caused obvious interference with the results.

SMI technology is based on high-resolution Doppler technology. We used Aplio series upmarket ultrasonic diagnostic equipment to build a high-density beamformer and real-time application platform and image the low-flow-velocity blood flow with a high frame rate. Traditional Doppler ultrasound uses filtering technology to eliminate noise and motion artifacts, resulting in the loss of low-speed blood flow information. SMI technology can identify the noise generated by blood flow and tissue movement, and it uses adaptive calculation to display real-time blood flow information, so that the low-speed blood flow signals can be separated from filtered clutter signals and displayed [27]. Recent studies have indicated that SMI is a simpler, safer, cheaper, and noninvasive technique and may facilitate the visualization of carotid artery IPN without the use of a contrast agent [7, 14]. However, no quantitative evaluations of IMVF signal in SMI have been performed, and the relationship between SMI findings and degree of enhancement on CEUS remain unclear. At present, there is a lack of multicenter, large-sample research. This study aims to provide a comprehensive and reliable conclusion on the consistency of SMI and CEUS in detecting IPN.

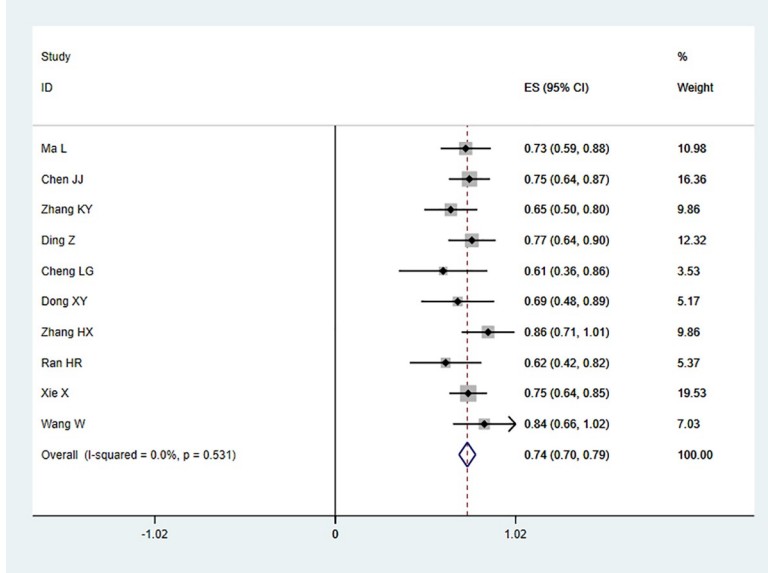

**Fig 3. Forest plots of Kappa index for superb microvascular imaging (SMI) in the detection of intraplaque neovascularization (IPN) comparable to contrast-enhanced ultrasonography (CEUS).**

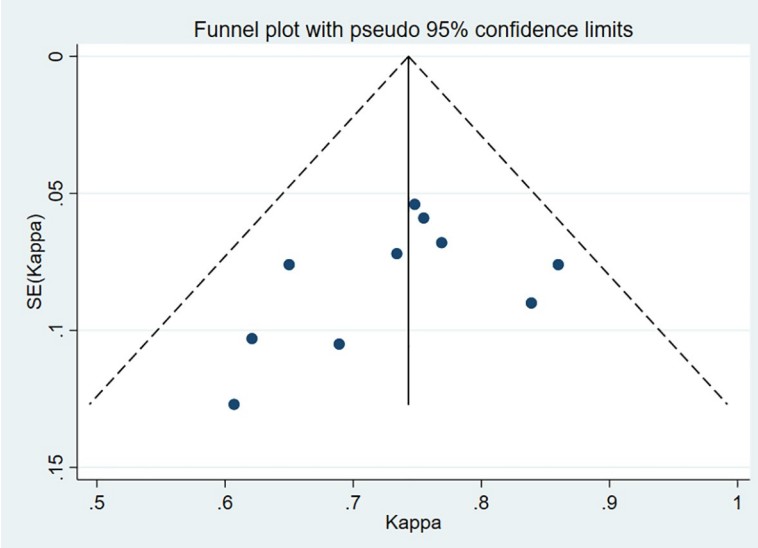

**Fig 4. Begger's funnel plot of publication bias on the pooled Kappa.** No publication bias was detected in this meta-analysis.

In the present meta-analysis, we systematically evaluated the technical performance and the consistency of SMI and CEUS in detecting IPN. Finally, 10 independent studies were included with a total of 608 carotid plaques assessed. The pooled summary Kappa index was 0.743 with statistical significance. Furthermore, our results indicated no direct evidence for publication bias. Taken together, consistent with previous studies, our findings strongly suggest that SMI and CEUS display a good consistency in detecting IPN of carotid plaque; that is, SMI ultrasound may be a promising alternative to CEUS for detecting IPN of carotid plaque.

However, this work does have several limitations. First, the included studies were mainly performed in China, which may have caused selection bias due to ethnicity factors. Although existing systematic reviews have suggested that the inclusion of articles published in Chinese alone would not affect the overall effect direction, the exclusion of publications in other languages may reduce the precision of the summary effect estimates [28]. However, many of the studies did not address whether the grades of SMI and CEUS interacted with blinding. Empirical evidence suggests that lack of blinding tends to cause overestimation of the treatment effect [29]. This indicates that even if bias is introduced by the lack of blinding, the true Kappa index would be even smaller than the results generated by this meta-analysis. Meta-analyses are retrospective studies, which may lead to subject selection bias. Therefore, more well-designed, large, multicenter, prospective, double-blind control studies should be conducted to validate these findings in future studies.

In conclusion, our meta-analysis suggests that SMI and CEUS display a good consistency in detecting IPN.

## Supporting information

**S1 Checklist. PRISMA 2009 checklist.**
(DOC)

## Acknowledgments

We would like to acknowledge the helpful comments on this paper received from our reviewers. We would also like to thank all our colleagues working in the Ultrasound Department of

the First Affiliated Hospital at Dalian Medical University. We thank LetPub (www.letpub. com) for its linguistic assistance during the preparation of this manuscript.

## Author Contributions

**Data curation:** Fang Yang.

**Investigation:** Fang Yang.

**Writing – review & editing:** Cong Wang.

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
