## [Decision Letter · Decision Letter 0]

18 May 2020

PONE-D-20-06672

Consistency of superb microvascular imaging and contrast enhanced ultrasonography in detecting intraplaque neovascularization: a meta-analysis

PLOS ONE

Dear Mr. Wang,

Thank you for submitting your manuscript to PLOS ONE. After careful consideration, we feel that it has merit but does not fully meet PLOS ONE’s publication criteria as it currently stands. Therefore, we invite you to submit a revised version of the manuscript that addresses the points raised during the review process.

We would appreciate receiving your revised manuscript by Jul 02 2020 11:59PM. To enhance the reproducibility of your results, we recommend that if applicable you deposit your laboratory protocols in protocols.io, where a protocol can be assigned its own identifier (DOI) such that it can be cited independently in the future. For instructions see: http://journals.plos.org/plosone/s/submission-guidelines#loc-laboratory-protocols

We look forward to receiving your revised manuscript.

Kind regards,

Dalin Tang

Academic Editor

PLOS ONE

Journal Requirements:

https://doi.org/10.1007/s13277-014-1815-2

In your revision ensure you cite all your sources (including your own works), and quote or rephrase any duplicated text outside the methods section. Further consideration is dependent on these concerns being addressed

4. Please include your tables as part of your main manuscript and remove the individual files. Please note that supplementary tables (should remain/ be uploaded) as separate "supporting information" files

Reviewers' comments:

Reviewer's Responses to Questions

**Comments to the Author**

1. Is the manuscript technically sound, and do the data support the conclusions?

Reviewer #1: Yes

Reviewer #2: Partly

2. Has the statistical analysis been performed appropriately and rigorously? 

Reviewer #1: Yes

Reviewer #2: Yes

3. Have the authors made all data underlying the findings in their manuscript fully available?

Reviewer #1: Yes

Reviewer #2: Yes

4. Is the manuscript presented in an intelligible fashion and written in standard English?

Reviewer #1: No

Reviewer #2: No

5. Review Comments to the Author

Reviewer #1: The ms. by Yang et al. is aimed at demonstrating that superb microvascular imaging and contrast enhanced ultrasonography are comparable in detecting intraplaque neovascularization. The authors performed a thorough literature research. Findings are clinical relevant and may pave the way for clinical application.

Overall, the finding is convincing and the manuscript well-written. However, method description and language are not always optimal.

My concerns are:

1) Did all studies use the same device from one manufacturer? What’s the potential impact on conclusion?

2) Did the interrater-reliability studies include especially concerning about the grading?

3) Is it possible to examine the plaque in the same plane comparing both methods?

4) Authors should make appropriate changes to the language in standard English.

Reviewer #2: The authors indicated that SMI would be a promising alternative to CEUS for detecting IPN of carotid plaque by meta-analysis, which seemed to be significant and useful to evaluate the stability of atherosclerosis plaque and predict the risk of cerebrovascular diseases becauses of its noninvasive, safe and convenient characteristics.

However, there are some problems and issues:

First, all the included 10 literatures were published in Chinese journals. How do the authors control the publication bias if there is only one kind of language literatures? Actually, there may be some international literatures which would meet the inculsion criteria. For example if such papers were published in one language neighter in chinese nor in English, but met the inclusion criteria.

Second, in the 10 literatures, some of the results were given double-blind evaluation by different doctors, others were not, how about the rate of deviation and heterogeneity caused by such issue?

Third, in the discussion part, though the authors indicate the limitations of this meta-analysis, there are no detailed measures to salvage them. The authors should address the objective reasons of the questions, and give some suggestions for next work.

In addition, there are some inappropriate sentences and words in the manuscript. Such as, in the first sentence of the Introduction part: “In today’s society, ….to be younger” and in the seventh sentence of this paragraph: “Several studies had…comparable to CEUS”, and so on. It seemed the English expression was incorrect. Please make serious changes of the language by someone with expertise in technical English editing before resubmitting.

6. PLOS authors have the option to publish the peer review history of their article (what does this mean?). If published, this will include your full peer review and any attached files.

Reviewer #1: No

Reviewer #2: No

---

## [Author Response · Author response to Decision Letter 0]

5 Jun 2020

Journal Requirements:

1.Please ensure that your manuscript meets PLOS ONE's style requirements, including those for file naming.

OK.

2.We noticed you have some minor occurrence of overlapping text with the following previous publication(s), which needs to be addressed:

Appropriate modifications have been made.

3.PLOS requires an ORCID iD:

OK.

4.Please include your tables as part of your main manuscript and remove the individual files. Please note that supplementary tables (should remain/ be uploaded) as separate "supporting information" files.

OK.

5.Please include captions for your Supporting Information files at the end of your manuscript, and update any in-text citations to match accordingly.

OK.

Reviewer #1: 

1)Did all studies use the same device from one manufacturer? What’s the potential impact on conclusion?

Yes, superb microvascular imaging(SMI; Toshiba Medical Systems Corporation, Tochigi, Japan).

All devices are from Toshiba aplio series color ultrasonic diagnostic instrument, which may reduce sources of heterogeneity.

2)Did the interrater-reliability studies include especially concerning about the grading?

No, the main statistical indicator is Kappa index. Risk of bias within studies was assessed through MINORS.

3)Is it possible to examine the plaque in the same plane comparing both methods?

Not always. Ultrasound usually scans in multiple sections, and takes the section with the most blood flow.

4) Authors should make appropriate changes to the language in standard English.

OK.

Reviewer #2:

First, all the included 10 literatures were published in Chinese journals. How do the authors control the publication bias if there is only one kind of language literatures? Actually, there may be some international literatures which would meet the inculsion criteria. For example if such papers were published in one language neighter in chinese nor in English, but met the inclusion criteria.

We have searched PubMed, Web of Science, Cochrane Library, and CBM databases in fact, but only the 10 literatures are eligible. The Begg’s funnel plot and Egger’s test were applied to assess the publication bias. We have add descriptionin ‘Although existing systematic reviews have suggested that the inclusion of articles published in Chinese alone would not affect the overall effect direction, the exclusion of publications in other languages may reduce the precision of the summary effect Estimates’ in the discussion part. 

Second, in the 10 literatures, some of the results were given double-blind evaluation by different doctors, others were not, how about the rate of deviation and heterogeneity caused by such issue?

Empirical evidence suggests that lack of blinding tends to lead to overestimation of the treatment effect. This implies that even if bias is introduced by the lack of blinding, the true Kappa index would be even smaller than the results generated by this systematic review.

Third, in the discussion part, though the authors indicate the limitations of this meta-analysis, there are no detailed measures to salvage them. The authors should address the objective reasons of the questions, and give some suggestions for next work.

OK.

In addition, there are some inappropriate sentences and words in the manuscript. Such as, in the first sentence of the Introduction part: “In today’s society, ….to be younger” and in the seventh sentence of this paragraph: “Several studies had…comparable to CEUS”, and so on. It seemed the English expression was incorrect. Please make serious changes of the language by someone with expertise in technical English editing before resubmitting.

OK.

---

## [Decision Letter · Decision Letter 1]

2 Jul 2020

PONE-D-20-06672R1

Consistency of superb microvascular imaging and contrast-enhanced ultrasonography in detection of intraplaque neovascularization: A meta-analysis

PLOS ONE

Dear Dr. Wang,

Thank you for submitting your manuscript to PLOS ONE. After careful consideration, we feel that it has merit but does not fully meet PLOS ONE’s publication criteria as it currently stands. Therefore, we invite you to submit a revised version of the manuscript that addresses the points raised during the review process.

We look forward to receiving your revised manuscript.

Kind regards,

Dalin Tang

Academic Editor

PLOS ONE

Reviewers' comments:

Reviewer's Responses to Questions

**Comments to the Author**

1. If the authors have adequately addressed your comments raised in a previous round of review and you feel that this manuscript is now acceptable for publication, you may indicate that here to bypass the “Comments to the Author” section, enter your conflict of interest statement in the “Confidential to Editor” section, and submit your "Accept" recommendation.

Reviewer #1: All comments have been addressed

Reviewer #2: All comments have been addressed

2. Is the manuscript technically sound, and do the data support the conclusions?

Reviewer #1: Yes

Reviewer #2: Yes

3. Has the statistical analysis been performed appropriately and rigorously? 

Reviewer #1: Yes

Reviewer #2: Yes

4. Have the authors made all data underlying the findings in their manuscript fully available?

Reviewer #1: Yes

Reviewer #2: Yes

5. Is the manuscript presented in an intelligible fashion and written in standard English?

Reviewer #1: Yes

Reviewer #2: Yes

6. Review Comments to the Author

Reviewer #1: This manuscript has been improved. The authors have addressed all my comments. I don't have further comments.

Reviewer #2: The authors have made lots of modification of the manuscript according to the comments.

In the Discussion part about limitations some suggestions should be mentioned for next work.

Please make further careful polish of language.

7. PLOS authors have the option to publish the peer review history of their article (what does this mean?). If published, this will include your full peer review and any attached files.

Reviewer #1: No

Reviewer #2: No

---

## [Author Response · Author response to Decision Letter 1]

4 Jul 2020

1.In the Discussion part about limitations some suggestions should be mentioned for next work.

Answer:OK.--line160-162.

2.Please make further careful polish of language.

Answer:OK.

3.While revising your submission, please upload your figure files to the Preflight Analysis and Conversion Engine (PACE) digital diagnostic tool.

Answer:OK.

---

## [Decision Letter · Decision Letter 2]

14 Jul 2020

Consistency of superb microvascular imaging and contrast-enhanced ultrasonography in detection of intraplaque neovascularization: A meta-analysis

PONE-D-20-06672R2

Dear Dr. Wang,

We’re pleased to inform you that your manuscript has been judged scientifically suitable for publication and will be formally accepted for publication once it meets all outstanding technical requirements.

Kind regards,

Dalin Tang

Academic Editor

PLOS ONE

Additional Editor Comments (optional):

Reviewers' comments:

Reviewer's Responses to Questions

**Comments to the Author**

1. If the authors have adequately addressed your comments raised in a previous round of review and you feel that this manuscript is now acceptable for publication, you may indicate that here to bypass the “Comments to the Author” section, enter your conflict of interest statement in the “Confidential to Editor” section, and submit your "Accept" recommendation.

Reviewer #2: All comments have been addressed

2. Is the manuscript technically sound, and do the data support the conclusions?

Reviewer #2: Yes

3. Has the statistical analysis been performed appropriately and rigorously? 

Reviewer #2: Yes

4. Have the authors made all data underlying the findings in their manuscript fully available?

Reviewer #2: Yes

5. Is the manuscript presented in an intelligible fashion and written in standard English?

Reviewer #2: Yes

6. Review Comments to the Author

Reviewer #2: The authors addressed all the concerns.

They added some limitations of this study in the discussion part and analyzed them well.

7. PLOS authors have the option to publish the peer review history of their article (what does this mean?). If published, this will include your full peer review and any attached files.

Reviewer #2: No

---

## [Editor Report · Acceptance letter]

16 Jul 2020

PONE-D-20-06672R2 

Consistency of superb microvascular imaging and contrast-enhanced ultrasonography in detection of intraplaque neovascularization: A meta-analysis 

Dear Dr. Wang:

I'm pleased to inform you that your manuscript has been deemed suitable for publication in PLOS ONE. Congratulations! Your manuscript is now with our production department. 

Kind regards, 

on behalf of

Professor Dalin Tang 

Academic Editor

PLOS ONE